# Thinking in pictures in everyday life situations among autistic adults

**Clara Bled** [1¤]*, **Quentin Guillon** [1], **Isabelle Soulières**[2], **Lucie Bouvet**[1]

**1** Laboratoire CERPPS-E.A. 7411, Université Toulouse Jean Jaurès, Toulouse, France, **2** Psychology Department, C.P. 8888 Succursale Centre-Ville, Université du Québec à Montréal, Montreal, Canada

¤ Current address: Laboratoire CERPPS, Université Toulouse Jean Jaurès, Toulouse, France
* clara.bled@univ-tlse2.fr

**Data Availability Statement:** All relevant data are within the manuscript and its Supporting Information files.

**Funding:** This work was funded by the French National Research Agency (ANR-19-CE28-0012).

## Abstract

Autistic individuals are often described as thinking in pictures. The aim of this study was to investigate the phenomenological characteristics of mental representations and inner experiences of autistic individuals. A total of 39 autistic adults and 80 control adults answered an online questionnaire. Autistic participants reported a more frequent use of visual mental representations than controls for different types of everyday situations. Moreover, autistic individuals defined their visual mental representations as more detailed than control participants. Furthermore, when describing their inner experiences, autistic participants used perceptive visual themes whereas control participants relied more on the description of events and memories. Our results support the hypothesis that some autistic individuals indeed "think in pictures". We discuss the impact of such a visual way of thinking in daily life.

## Introduction

According to the Diagnostic and Statistical manual of Mental disorders, fifth edition (DSM-5), autism is a neurodevelopmental disorder characterized by impairments in social communication and interaction, as well as restricted and repetitive patterns of behaviors or interests [1]. Autism therefore entails behavioral but also cognitive atypicalities. Autistic individuals are often described as "thinking in pictures". This assumption mainly originates from the eponymous book by Temple Grandin, *Thinking In Pictures*. In this book she reports that "*when I think about abstract concepts I use visual images*", "*thinking in language and words is alien to me. I think in pictures*" [2].

Kunda and Goel [3] hypothesized that some performance of autistic individuals can be explained by this tendency to think in pictures and the use of visual strategies with visual mental representations. For instance, a greater activation of the visual cortex has been observed throughout different tasks [4–6]. Moreover, autistic individuals also use visual strategies and mental representations when performing tasks of language comprehension and verbal reasoning [7, 8].

Another example of the use of visual strategies in autistic individuals and its behavioral outcome is the smaller word length effect during a serial recall task. This result suggests that autistic individuals rely more on the visual representation of words than controls [9]. In addition,

The funders had no role in study design, data collection and analysis, decision to publish, or preparation of the manuscript.

**Competing interests:** The authors have declared that no competing interests exist.

in a recent literature review, Williams et al. [10] found that articulatory suppression, which prevents the use of verbal strategies, had no impact on the performances of autistic individuals. The authors hypothesized that cognition in autistic individuals may rely less on internal language or "inner speech" than in controls, as they use alternative, i.e. visual, strategies [10].

Although these studies have characterized the use of visual strategies in autistic individuals or the ability of autistic individuals to use / manipulate visual mental representations with laboratory tasks [11–14], very few studies have documented the phenomenology of mental representations or explored the inner experiences of autistic individuals in daily life.

Cognitive theories concentrate on mental processes of conceptual thinking whereas phenomenology pays attention to the subjective processes displayed "within" the individual's contact with the world. Phenomenology finds its roots in philosophy. It is an approach that focuses on the form and structure of conscious experience with descriptions of how we think (mode/form), what we think (content) and why we think like we do (insight/understanding). This subject-oriented or 1$^{st}$ person approach may not only be relevant in a diagnostic process of autism [15] but may also shed light on dimensions of cognitive experience that are lacking in current research [16].

The investigation of the subjective experience can provide a unique possibility to explore how people with autism experience themselves, others, and the world [17]. There are indeed several theoretical accounts of how phenomenological concepts can inform our understanding of subjective experience in autism. Some phenomenological theoretical papers questioned the theory of mind and the capacity of autistic individuals to understand their own mind [18] or the capacity of intersubjectivity (e.g., a pre-reflective embodied relationship of self and other in an emergent bipersonal field) in autism [19]. Other phenomenological clinical studies explored the sensory atypicalities [20] or subjective experience of time in autism [21]. While these studies have focused primarily on the content of autistic individuals' thoughts (i.e. what they think), the current study focuses instead on the form of mental representations (i.e. how they think) as well as their basic phenomenological characteristics.

In 1993, Hurlburt developed a method called the Descriptive Experience Sampling (DES) method, to explore people's inner experience [22]. This method consisted in wearing, for several days in a row, a small device that beeped at random intervals. When they heard a beep, participants had to "freeze" the content of their awareness and write it down on paper. Four major categories of inner experience were identified: verbal inner experiences (or inner speech), visual images (or visual mental representations), unsymbolized thinking (or what they described as "pure thought") and feelings (or emotional experiences). Using this method with three adults with Asperger syndrome, Hurlburt et al. were the first to explore the form of autistic individuals' thought in a systematic way in their daily lives [23]. They found that these three individuals reported thoughts primarily or solely in the form of visual images. More recently, Hare et al. also noted the importance of visual thinking in autism in everyday life [24]. Using a similar approach known as experience sampling methodology (ESM) where participants are invited several times a day for several days in a row to answer a series of questions when prompted by a signal, they found that the most common form of thought among Asperger participants was images reported in 40% of the prompts, followed by inner speech reported in 37.5% of the prompts. In contrast, control participants mainly reported thoughts in the form of inner speech (68.4% vs. 19.1% for thoughts as images).

In the current study, we investigated the form of mental representations and inner experiences of autistic individuals in everyday life using an online questionnaire. The use of an online questionnaire makes it more accessible and allows more people to participate. Based on the thinking in pictures account by Kunda and Goel [3] as well as reports by Hurlburt et al. [23] and Hare et al. [24], we hypothesized that autistic participants would report a greater use

of visual mental representations than control, non-autistic, participants in everyday life situations. To further explore and characterize the inner experience of these visual mental representations, participants answered questions about basic phenomenological characteristics, including their duration, level of detail and manipulation. We also explored inner experiences with an open-ended question to assess the use of visual mental representations via a textual analysis of their written production.

## Material and method

The study was approved by the ethics committee of the University of Toulouse (CERNI 2018–112) and all participants read an information leaflet with explanations concerning the study and gave their written consent before beginning the questionnaire.

### 1. Participants

Participants were recruited by posting an advertisement (containing a direct link to the questionnaire) in local clinical centers and associations for autistic adults, on social networks and forums about autism. To be included in the analyses, participants had to be over 18 and had to provide enough details about their diagnosis (specifying the age at diagnosis, institution of diagnosis and the diagnostic tool that was used). A total of 79 autistic participants responded to the questionnaire, of whom 71 were over 18 and provided enough details about their diagnosis. Of these, 32 did not fully complete the questionnaire and were excluded. Given the fixed order of questions, the pattern of missing data was not random and we therefore decided to exclude all protocols that were not fully complete. In total, 39 adults (11 men, 28 women) with a formal diagnosis of autism based on DSM criteria (either DSM-IV or DSM 5) were included in the analyses (age range 18 to 62, mean = 33.5, SD = 10.6). Among them, 20 were assessed with the Autism Diagnostic Observation Schedule (ADOS 2) [25] and/or the Autism Diagnostic Interview (ADI) [26] and the remaining 19 with other standardized tools (e.g. Australian Scale for Asperger Syndrome (ASAS-R) [27], Ritvo Autism Asperger Diagnostic Scale (RAADS-R) [28], Autism spectrum Quotient (AQ) [29]). All diagnoses were confirmed by a medical doctor in a specialized autism resource center (CRA) (45.2%), in a private practice (41.9%) or in a hospital (12.9%). Most of our respondents received a late diagnosis (age range at diagnosis 4 to 56 years, mean = 29.5, SD = 12.8). Out of 39 autistic participants, 12 reported a depressive episode in the past and 1 a Post-traumatic Stress Disorder (PTSD), 6 declared having an anxiety disorder, 9 an Attention Deficit Hyperactivity Disorder (ADHD), 3 dyspraxia, 1 dyslexia, 1 dyscalculia and 2 a High Intellectual Potential (HIP).

281 control participants also responded to the questionnaire, but two of them were under 18, 175 did not fully complete the questionnaire, 24 had close family members with autism and so were removed from analysis. Thus, a total of 80 comparison participants (24 men, 56 women) were included (age range 18 to 69, mean = 35.9, SD = 12.6). Most control participants had no psychiatric history but 8 of them reported a depressive episode in the past, 5 declared having a High Intellectual Potential (HIP), 6 an Attention Deficit Hyperactivity Disorder (ADHD) and 1 a visuospatial dyspraxia.

The two groups did not significantly differ in terms of age (t(117) = 1.06, $p$ = .292), education level ($\chi^2$(5) = 9.81, p = .081) and gender (70% of women in the control group, 71.8% in the autistic group, $\chi^2$(1) = .04, p = .840).

### 2. General procedure

The questionnaire was created using *LimeSurvey©*, and securely hosted by the University Toulouse–Jean Jaurès.

The questionnaire contained a total of 66 items and took approximately 20 minutes to complete. For the purpose of this study, 11 of these items were analyzed. The original French version of these 11 items is available in supplementary data. These items pertained to three blocks of questions presented in the same fixed order for all participants. First, participants answered three questions related to the phenomenological characteristics of their visual representations. Then, they were asked an open question about what comes to their mind when they hear the name of a city they know well. The third block consisted of seven questions related to the use of visual mental representations in everyday life situations.

Participants could, at any time, stop the questionnaire and erase their answers or continue the questionnaire later. They also had the possibility of backtracking if they wished to correct some of their answers.

## 3. Measures

The 11 items of the questionnaire were created based on pre-existing mental representation questionnaires or interviews [23, 30, 31].

**Phenomenological characteristics of the visual mental representations.** These 3 items assessed how participants subjectively experience their visual mental representations. We were interested in the duration of mental images, the level of detail of mental images and the ability to manipulate mental images. Prior to answering these questions, participants were asked to mentally visualize an object or a situation of their choice and answer the questions in relation to the triggered internal image. There were three possible answers for each question. The three items are as follows:

1. *"Is this internal image **detailed** or **blurry** or **sometimes blurry and sometimes detailed***?

2. *Is this internal image **persistent** or **of short duration** or **sometimes persistent and sometimes of short duration***?

3. *Do you manage to manipulate this internal image (e.g. change point of view, zoom in, etc.)?* ***Always*** *or* ***never*** *or* ***sometimes***?*"

For each question, the last two responses, indicating poorer visual mental representations, were grouped together for analysis.

**Open question.** We asked an open question in order to assess the inner experiences of the participants qualitatively.

"*Describe what comes to your mind when you hear the name of a city you have already been to, such as Toulouse.*" (Note that Toulouse is a city well known to all participants.)

Participants had to type their answer in a box reserved for this purpose.

**Use of visual mental representations.** Participants indicated if they usually use images and/or words for 7 different "everyday" situations: recollection, problem solving, anticipation, decision making, planning, comprehension and memorization. The 7 items are as follows:

1. *"When you remember something (that you have read, seen or heard), do you tend to use images, words or both*?

2. *To understand something that is being explained to you (e.g. instructions), do you tend to use images, words or both*?

3. *When you anticipate an upcoming event (e.g. if you have to go somewhere), do you tend to use images, words or both*?

4. *When planning activities (e.g. if you are thinking about how to organize your day), do you tend to use images, words or both*?

5. *When you face a problem and cannot find a solution, do you tend to use images, words or both?*

6. *When you have to make a decision (e.g. the best way to go somewhere), do you tend to use images, words or both?*

7. *When you have to memorize something (e.g. a shopping list, a phone number, etc.) do you tend to use images, words or both?"*

For each situation, the response "words only" was scored 0, "images and words" was scored 1, and "images only" was scored 2.

## 4. Data analysis

To verify the covariance of scores on different items and justify the use of a global score for the use of visual mental representations, Cronbach's alpha was used. To account for the ordered nature of the variable, a Mann-Whitney *U* test (one-tailed) was used to test for a between-group difference in the use of visual mental representations. Statistical significance was set to a p-value of 0.05. Interquartile ranges (IQR) are reported as a range between the $1^{st}$ and $3^{rd}$ quartiles to convey information about the asymmetrical/symmetrical distribution of scores around the median. In addition, we calculated a Glass rank biserial correlation coefficient (*r*g) to assess the effect size of the difference. For between-group comparisons of response frequencies with respect to the phenomenological characteristics, we used the Pearson chi-square test. Given the exploratory nature of these analyses, we did not correct for multiple comparisons, limiting the risk of false negative results. Statistical analyses were performed using *SPSS* V23.0 [32].

A lexicometric analysis was performed on participants' responses to the open question. *Iramuteq* software [33] was used for frequency analysis, similarity analysis and the Descending Hierarchical Classification (DHC) method. The statistical procedure of DHC was chosen for the cluster method of lexical classes [34], allowing us to show the distribution of lexemes (i.e. units of meaning) associated with relevant areas of textual content across the overall corpus of data.

## Results

### 1. Use of visual mental representations

Regarding the use of visual mental representations, the seven questions had a satisfactory covariance ($\alpha$ = .81), thus a score for the use of visual mental representations was computed with the sum of the seven scores. The higher the score (max = 14), the more frequent and extended the use of visual mental representations in everyday life.

The median score for the use of visual mental representations was 10 (IQR = 7.5–13) for the autistic group and 8 (IQR = 6–10) for the control group. The frequency of use of visual mental representations was significantly higher in autistic individuals than in controls (U = 1031.1, *p* = .001 one-tailed) (see Fig 1). The Glass rank biserial correlation coefficient *r*g = .34 indicated a medium effect size.

At the individual level, 5 control participants (6.2%) reported using images only (indicated by a score of 14) and 1 control participant reported using words only (indicated by a score of 0). In contrast, 6 autistic participants (15%) reported using images only and none reported using words only.

As there were a large number of women in our participants, we explored a potential gender effect in the use of visual mental representations. In the control group, the median score for the use of visual mental representations was 7.5 (IQR = 7–11.25) for men and 8 (IQR = 5–10)

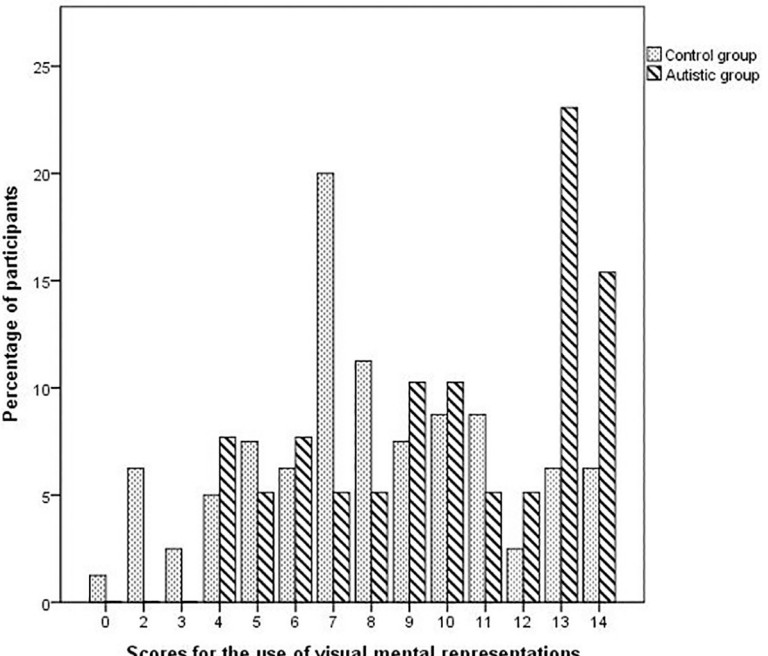

**Fig 1. Distribution of scores for the use of visual mental representations in autistic individuals and in controls.**
The higher the score, the more frequent and extended the use of visual mental representations.

for women, this difference was not significant (U = 589.5, $p$ = .383 two-tailed). In the autistic group, the median score for the use of visual mental representations was 9 (IQR = 7.5–12.5) for men and 11 (IQR = 7.75–13) for women, this difference was also not significant (U = 124.5, $p$ = .363 two-tailed).

## 2. Phenomenological characteristics of visual mental representations

One control participant who reported having no visual mental representations was excluded from the following analyses.

The use of visual mental representations was more frequent in the autistic group but we also wanted to explore whether the basic characteristics of these representations were experienced differently across groups. Visual mental representations were more frequently reported as detailed among autistic individuals, with 66.7% of autistic participants reporting *detailed* mental images versus 43.8% of control participants ($\chi^2(1)$ = 5.51, $p$ = .019). Visual mental representations were not more frequently reported as persistent among autistic individuals (35.9% vs. 20%), $\chi^2(1)$ = 3.51, $p$ = .051. Regarding the ability to manipulate mental images, the proportion of participants who reported always being able to manipulate their images did not differ between groups either (56.4% vs. 46.3%), $\chi^2(1)$ = 1.08, $p$ = .298.

## 3. Open question

According to the frequency analysis, the answers of autistic individuals were as detailed as those of control participants with the same degree of use of descriptive adjectives (relative frequency of use of adjectives was 59.65 for controls and 53.83 for autistic individuals). The most frequently reported forms (or words) by the controls were: "ville" (*city*) (33 occurrences), "souvenirs" (*memories*) (26 occurrences) and "lieu" (*place*) (21 occurrences). The most frequently reported forms by autistic individuals were: "image" (*image*) (16 occurrences), "voir" (*see*) (13

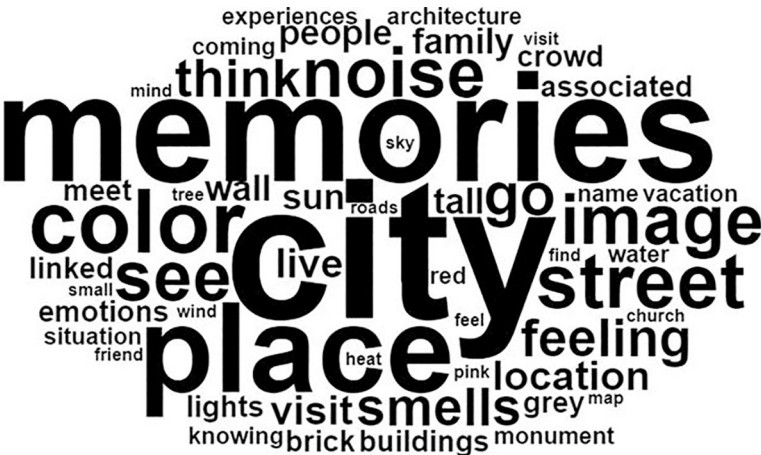

**Fig 2. Word cloud for the control group** *(the original French word cloud is available in supplementary data).*

occurrences), "ville" (*city*) (13 occurrences) and "visualiser" (*visualize*) (9 occurrences) (see Figs 2 and 3).

A similarity analysis was also performed. The similarity index corresponds to the co-occurrence (simultaneous appearance) of forms in the same text segment (the higher the index, the more present the link between concepts). In controls, the word "ville" (*city*) was the most frequently used and was often associated with "lieu" (*place*) (with a co-occurrence index of 8). "Lieu" *(place)* was frequently associated with "souvenirs" (*memories*) (co-occurrence of 7), which was associated with "personnes" (*people*) (co-occurrence of 5) and "sensations" (*sensations*) (co-occurrence of 5). In autistic individuals, "image" (*image*) was the central word and was associated, in order of importance, with "ville" (*city*), "voir" (*see*), "odeurs" (*smells*) and "couleur" (*color*) (co-occurrence indices were respectively: 6, 6, 4 and 3).

These analyses suggest the use of different lexical fields in the control group relative to the autistic group. To go further, we then carried out a Descending Hierarchical Classification (DHC), with answers from all participants, in order to group together the common lexical fields and to highlight the different themes appearing in the individuals' answers (see Fig 4).

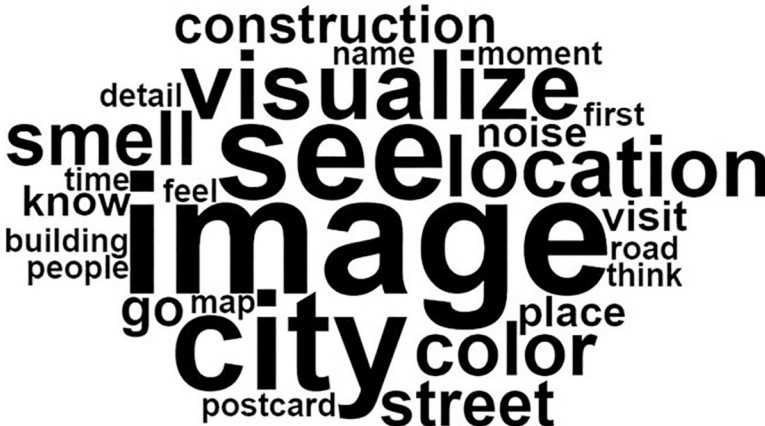

**Fig 3. Word cloud for the autistic group** *(the original French word cloud is available in supplementary data).*

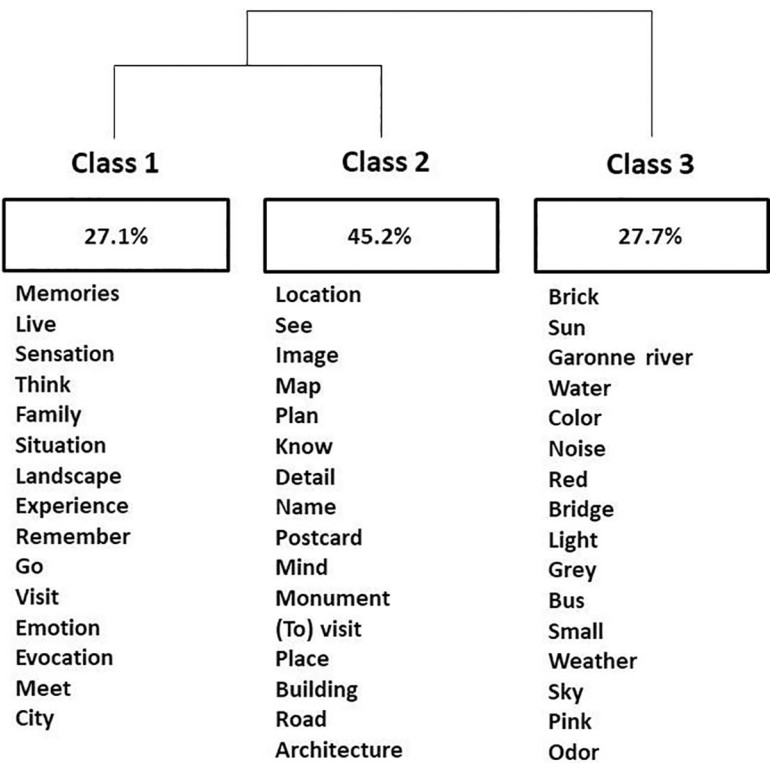

**Fig 4. Descending hierarchical classification of lexical classes.** Class 1 = recall class, class 2 = visual representation class, class 3 = perceptive details class (the original French dendrogram is available in supplementary data).

The DHC was able to classify 94.51% of the text segments, and distinguished 3 classes among these segments. Class 1 contained 27.1% of the data, the second class 45.2% and the third class 27.7%. Two subsets of classes stand out: classes 1 and 2 on the one hand and class 3 on the other. The dendrogram provides a list of the forms most closely associated with each class (see Fig 4).

Class 3 seems related to the theme of perception with many perceptive details reported by participants (sun, water, color, noises, light, etc.). We named class 3 the "perceptive details class".

Class 2 is more specifically associated with representative and visual terms (see, image, map, plan, etc.). Here are some examples of a characteristic text segment associated with class 2: "Je visualise précisément, comme des cartes postales, des images de lieux que je connais [. . .]" (*I visualize precisely, like postcards, images of places I know [. . .]*). "[. . .] un plan d'ensemble d'un lieu emblématique [. . .] l'architecture globale et l'allure des bâtiments. Un point de vue souvent impersonnel, un peu comme une image de carte postale." *([. . .] an overall map of an emblematic place [. . .] the overall architecture and the shape of the buildings. An often impersonal point of view, a bit like a postcard image).* We named class 2 the "visual representation class".

Class 1 was associated with the theme of description of events and memories (memories, live, sensations, family, etc.). An example of a characteristic text segment associated with class 1 is: "[. . .] les villes ressortiront plus pour les rencontres et les expériences dans ces villes, et donc les souvenirs et sensations plus que la ville en elle-même" (*[. . .]cities will stand out more for the encounters and experiences in these cities, so the memories and sensations more than the city itself*). We named class 1 the "recall class".

Text segments associated with the "recall class" were significantly more frequently reported by control participants ($\chi^2$ = 13.33, $p$ < .001), whereas text segments associated with the "visual representation class" were significantly more frequently reported by autistic participants ($\chi^2$ = 10.86, $p$ < .001). The "perceptive details class" was not specifically associated with either group.

Altogether, these lexicometric analyses indicate that control participants were more likely to use the themes of recall and memories, while autistic participants tended to use visual representation terms with, for example, the word "image" (*image*) which was significantly associated with the "visual representation class" ($\chi^2$ = 14.11, $p$ < .001).

## Discussion

The aim of this study was to investigate the use of visual mental representations and explore their phenomenological characteristics in autistic individuals.

Consistent with our hypothesis, autistic participants as a group reported a more frequent and extended use of visual mental representations for different types of everyday life situations (recollection, problem solving, anticipation, decision making, planning, comprehension and memorization) than control participants. This more frequent use of visual representations is consistent with what Hurlburt et al. [23] reported on the first attempt to investigate the inner experience of three participants with Asperger syndrome. Our results are also in agreement with Hare et al.'s findings that the most frequent forms of thoughts in autistic individuals are images [24]. A possible explanation for this greater use of visual representations is that autistic cognition relies more on an enhanced visual, perceptual functioning [35–37]. Though these questions remain open, there might be a link between the particular sensory / perceptual functioning in autism and the use of visual strategies. A greater use of visual mental imagery in participants with autism is mechanistically accompanied by a lower use of words / verbal representations in comparison to control participants. Indeed, typical individuals more frequently reported thoughts as if they were talking to themselves [24], a form of inner speech. Therefore, another possibility is that autistic individuals may rely relatively less on words and consequently more on visual representations because of an impaired "inner speech" (see Williams et al.'s review on inner speech in autism [10]). In this study, we did not collect information about inner speech. This latter possibility thus remains to be investigated in future studies.

Our results extend prior reports by suggesting that autistic individuals differed from control individuals not only in the frequency of use of visual mental representations but also in some phenomenological characteristics. Autistic participants were more likely than control participants to define their mental images as detailed. Previous studies also revealed that autistic individuals show enhanced performances in both low- and high-level visual perception tasks and are generally better at detecting purely perceptual changes [35–37]. Given that perception and mental visual representations activate the same neural networks and rely on the same representations [38], it is very likely that the attention to visual details in autistic individuals at the perceptual level is mirrored at the level of visual mental representations. Regarding the persistence and the manipulation of the visual mental representations, findings were not clear and would require further investigation.

Finally, inspired by preexisting interviews on inner representations [23, 30], we investigated how participants described their inner experiences when reporting what came to their mind when they heard the name of a city. We found that autistic and control participants did not use the same lexical field in their answers. Autistic participants tended to use the themes of visual representations and perception whereas control participants mostly used the themes of

recall and memories, a result consistent with a thinking in picture hypothesis manifested as a greater use of visual mental representations. Alternatively, autistic individuals may not instinctively report memories possibly due to autobiographical memory difficulties. Studies have shown that autistic individuals generate fewer specific autobiographical memories than typical adults and take significantly longer to do so [39], which could also explain our results.

Overall these findings are consistent with a thinking in pictures cognitive style in autism [3] and extend this in some way to everyday life situations. However, it is also important to recognize the variability in responses across the autistic group. Although a predominance of visual mental representations is observed at the group level, several participants reported a limited use of visual mental representations in everyday life situations. Characterizing this variability and exploring inter-individual differences will be important for future studies [40]. Further investigation will also be needed to understand how a thinking in pictures cognitive style might interfere with social cognition in autism. One of the challenges that is potentially posed by "thinking in pictures" is that pictures are detailed and context-specific. Thinking in pictures could thus make social abstractions more difficult.

Another important avenue for future research will be to study the potential implications of such a thinking in pictures cognitive style. For example, Höffler et al. demonstrated that high visualizers outperform low visualizers when learning from pictures and that cognitive style influences learning preferences [41]. Enhanced performances are commonly reported among autistic individuals in tasks in which the information is presented in a visual, structured and simultaneous manner, such as Raven's Progressive Matrices or the Wechsler Block Design task [42–44]. In autism, the visuo-spatial presentation of structured information is thus likely to support learning. A thinking in pictures cognitive style might also have clinical implications. Mental imagery plays a pivotal role in the onset and maintenance of mental disorders, and higher prevalence rates of mental disorders, such as anxiety, depression or post-traumatic stress disorder, are increasingly documented in the autistic population [45]. Although the exact nature of a potential link between a visual cognitive style and the onset/maintenance of mental disorders has not been studied yet, Ozsivadjian et al. found that autistic children have a greater propensity towards experiencing anxious imagery when compared to their non-autistic counterparts, even if they were not anxious [46].

## Limitations

Our study has several limitations. First and foremost, the relatively small number of autistic participants included clearly limits the generalization of the results. In the context of an online study, the selection of individuals and diagnosis confirmation are complex issues. We requested details about diagnosis and the diagnostic process (where the diagnosis was given, what diagnostic tools were used, etc.) and excluded participants who did not provide complete information, resulting in a smaller but higher quality sample. A relatively high attrition rate from initiation to survey completion was also observed, which poses the risk of sampling error. In particular, we cannot discard the possibility that the results apply only to a sub-group of individuals who were more inclined to complete the questionnaire because they do actually use visual strategies in daily life. A replication of these results in a larger and more representative sample is needed. Another limitation of our study is the rather high women to men ratio (2 to 3 women for 1 man). A high ratio of women is often reported in studies using online questionnaires [47]. Of note, the autistic and control groups were matched on gender (thus we have the same ratio in the autistic and control groups) and we did not find a significant gender effect in the use of visual mental representations in either group. Clearly, further studies will be needed to explore potential gender differences in the use of visual strategies in daily life.

Finally, although online questionnaires should allow for a relatively more extensive recruitment, they do not allow for an in-depth investigation of inner experiences as the DES or ESM methods can allow. Nevertheless, online questionnaires foster the participation of a wider variety of autistic individuals, including those less comfortable with face-to-face interviews or who are less willing to take part in research involving labor-intensive procedures such as the DES [23] or ESM [24] methods. We argue that these two approaches are complementary and will be important to further explore and characterize the form and phenomenological characteristics of mental representations in autistic individuals in daily life.

## Conclusion

In conclusion, this study suggests a greater use of visual mental representations in everyday life situations in autism as a group, a result consistent with a thinking in pictures account. Further investigations will be necessary to replicate the present findings in larger and more representative samples and with different methodologies, as well as to deepen the exploration of phenomenological characteristics, inter-individual variability and functional consequences in everyday life.

## Supporting information

**S1 Data.**
(CSV)

**S1 Appendix.**
(DOCX)

**S2 Appendix.**
(DOCX)

**S3 Appendix.**
(DOCX)

## Acknowledgments

We thank the participants for their time and commitment and the anonymous reviewers for their comments.

## Author Contributions

**Conceptualization:** Clara Bled, Lucie Bouvet.

**Data curation:** Clara Bled.

**Formal analysis:** Clara Bled.

**Funding acquisition:** Lucie Bouvet.

**Methodology:** Clara Bled, Quentin Guillon, Isabelle Soulières, Lucie Bouvet.

**Supervision:** Quentin Guillon, Isabelle Soulières.

**Validation:** Lucie Bouvet.

**Writing – original draft:** Clara Bled.

**Writing – review & editing:** Quentin Guillon, Isabelle Soulières, Lucie Bouvet.

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
