## [Decision Letter · Decision Letter 0]

21 Apr 2021

PONE-D-20-39247

Thinking in pictures in everyday life situations among autistic adults.

PLOS ONE

Dear Dr.  Bled,

Thank you for submitting your manuscript to PLOS ONE. After careful consideration, we feel that it has merit but does not fully meet PLOS ONE’s publication criteria as it currently stands. Therefore, we invite you to submit a revised version of the manuscript that addresses the points raised during the review process.  We apologise for the delay in getting back to you but the pressures associated with COVID have made it more difficult to find reviewers. Reviewer one has made  valuable suggestions for revising your manuscript. I have also read your manuscript  and agree with the points raised. The abstract should mention that the ASD sample consists of adults.  In the methods you need to provide more detail on the how the phenomenological  characteristic questions were triggered and how these questions were asked. In the results include measures  of dispersion  when describing group characteristics in addition to the median.  In the discussion expand upon potential implications of thinking in pictures including both benefits and disadvantages. As children use more imagery  than adults you could  think about the developmental changes with age in ASD.  Would the difference  between ASD participants and neurotypical controls possibly increase  with age? The limitations section needs to be expanded in a number of ways such as discussing the differences arising from using open-ended questions and the seven everyday situations questions. Finally check your manuscript for typographical errors like the following sentencer characterizing this variability and explore ( ing ) inter-individual differences. Address all points raised by Reviewer 1 and myself in your response.

Please submit your revised manuscript by 31/05/2021. If you will need more time than this to complete your revisions, please reply to this message or contact the journal office at plosone@plos.org. Please include the following items when submitting your revised manuscript:

We look forward to receiving your revised manuscript.

Kind regards,

Barbara Dritschel, PhD

Academic Editor

PLOS ONE

Journal Requirements:

2. Please provide additional details regarding participant consent. In the ethics statement in the Methods and online submission information, please ensure that you have specified what type you obtained (for instance, written or verbal, and if verbal, how it was documented and witnessed). For additional information about PLOS ONE ethical requirements for human subjects research, please refer to http://journals.plos.org/plosone/s/submission-guidelines#loc-human-subjects-research.

3. Peer review at PLOS ONE is not double-blinded (https://journals.plos.org/plosone/s/editorial-and-peer-review-process). For this reason, authors should include in the revised manuscript all the information removed for blind review, including the names of universities.

Reviewers' comments:

Reviewer's Responses to Questions

**Comments to the Author**

1. Is the manuscript technically sound, and do the data support the conclusions?

Reviewer #1: Yes

2. Has the statistical analysis been performed appropriately and rigorously? 

Reviewer #1: Yes

3. Have the authors made all data underlying the findings in their manuscript fully available?

Reviewer #1: Yes

4. Is the manuscript presented in an intelligible fashion and written in standard English?

Reviewer #1: Yes

5. Review Comments to the Author

Reviewer #1: Overall, this is an interesting and well-presented work. It represents a novel approach to identifying the use of mental representation among adults with Autism.

Based on the data collected, analytic procedures are appropriate and accurately described.

Discussion is consistent with findings generally. It would appear that the authors could expand upon the potential implications of thinking in pictures. For example, one of the challenges that is potentially posed by “thinking in pictures” is that pictures are detailed and specific to context, while words are abstractions from generic experience. It would seem that thinking in pictures would make social abstractions more challenging, and may have limited utility in changing from one context, in which contextual detail may be relevant, to another context in which abstracting the “gist” of past experience may be most useful. Thinking in pictures may be a hindrance to social cognition.

Specific Comments

It would be helpful to identify in the abstract if participants with autism were adults, as the comparison group is adults.

Attrition is relatively high from initiation to survey completion among people with Autism (45%) and controls (63%) and clearly represents a weakness. The included sample was also majority female (72%), which seems atypical for the larger population of adults with autism in which prevalence among males is higher than among females. Although both of these are touched upon in limitations, a more thorough examination of the potential implications of these limitations is needed.

It would help to clarify procedures if authors could identify what triggered the phenomenological characteristics questions. Were these follow up questions if participants identified images in the seven everyday situations? If they were follow up to the seven situations, were they triggered only for the “images only” response or for images and “images and words”?

Based on the provided survey questions, it is not clear how the phenomenological characteristics questions were asked. They are listed first, but it’s not clear what image is being referenced.

The open ended question references a relatively specific phenomenon, whereas the seven everyday situations are much more general in nature. Might this result in differences in the nature of how they are conceptualized?

Page 10 reports medians, which is consistent with a non-parametric analysis, but some report of dispersion (e.g. IQR) would also help to describe the groups.

6. PLOS authors have the option to publish the peer review history of their article (what does this mean?). If published, this will include your full peer review and any attached files.

Reviewer #1: **Yes: **Bryan P. McCormick

---

## [Author Response · Author response to Decision Letter 0]

28 Jun 2021

Editor:

1) The abstract should mention that the ASD sample consists of adults.

* Please see response #2 to the reviewer

2) In the methods you need to provide more detail on how the phenomenological characteristic questions were triggered and how these questions were asked.

* Please see responses #4 and #5 to the reviewer

3) In the results include measures of dispersion when describing group characteristics in addition to the median.

* Please see response #7 to the reviewer

4) In the discussion expand upon potential implications of thinking in pictures including both benefits and disadvantages. As children use more imagery than adults you could think about the developmental changes with age in ASD. Would the difference between ASD participants and neurotypical controls possibly increase with age?

* Please see response #1 to the reviewer.

5) The limitations section needs to be expanded in a number of ways such as discussing the differences arising from using open-ended questions and the seven everyday situations questions.

* Please see response #6 to the reviewer.

6) Finally check your manuscript for typographical errors like the following sentencer characterizing this variability and explore ( ing ) inter- individual differences.

* We have carefully checked the manuscript and hope that any errors have now been ironed out.

Reviewer:

1) Discussion is consistent with findings generally. It would appear that the authors could expand upon the potential implications of thinking in pictures. For example, one of the challenges that is potentially posed by “thinking in pictures” is that pictures are detailed and specific to context, while words are abstractions from generic experience. It would seem that thinking in pictures would make social abstractions more challenging, and may have limited utility in changing from one context, in which contextual detail may be relevant, to another context in which abstracting the “gist” of past experience may be most useful. Thinking in pictures may be a hindrance to social cognition.

* We have now broadened the potential implications of picture thinking, both in terms of its strengths in the context of learning, and in terms of its potential clinical implications, such as making autistic individuals potentially more vulnerable to mental health problems, e,g, anxiety or depression. The reviewer's suggestion is interesting in many ways and we have also included it to point out the need for future studies to investigate the link between picture thinking and social cognition difficulties in autism (see p.17).

2) It would be helpful to identify in the abstract if participants with autism were adults, as the comparison group is adults.

* We have now specified in the abstract that the autistic participants were adults.

3) Attrition is relatively high from initiation to survey completion among people with Autism (45%) and controls (63%) and clearly represents a weakness. The included sample was also majority female (72%), which seems atypical for the larger population of adults with autism in which prevalence among males is higher than among females. Although both of these are touched upon in limitations, a more thorough examination of the potential implications of these limitations is needed.

* We now further discuss the potential implications of these limitations, both in terms of sampling error and the need for further studies with larger, more representative samples. Please also note that we have reorganized the limitations by moving up the discussion of the sex ratio in the list of limitations, prior to the discussion regarding the use of an online questionnaire (see pages 17-18 in our revised manuscript).

4) It would help to clarify procedures if authors could identify what triggered the phenomenological characteristics questions. Were these follow up questions if participants identified images in the seven everyday situations? If they were follow up to the seven situations, were they triggered only for the “images only” response or for images and “images and words”?

* We apologize for this lack of clarity. Questions about the strategies used in everyday situations and questions about phenomenological characteristics were independent of each other (i.e. the phenomenological characteristics were not a follow up to the seven everyday situations). The phenomenological characteristics questions were asked before the seven everyday situations, in the order in which they are listed in the appendix (comment #5). We have now clarified this in the text.

* ‘’These [11] items pertained to 3 blocks of questions presented in the same fixed order for all participants. First, participants answered three questions related to the phenomenological characteristics of their visual representations. Then, they were asked an open question about what comes to their mind when they hear the name of a city they know well. The third block consisted of seven questions related to their use of visual mental representations in everyday life situations.’’(p. 7-8)

5) Based on the provided survey questions, it is not clear how the phenomenological characteristics questions were asked. They are listed first, but it’s not clear what image is being referenced.

* We apologize for this lack of clarity. Prior to answering these questions, participants were asked to generate a mental image of an object or a situation of their choice and to base their responses on this internal image. We have now clarified this in the text and amended the appendix accordingly.

6)The open ended question references a relatively specific phenomenon, whereas the seven everyday situations are much more general in nature. Might this result in differences in the nature of how they are conceptualized?

* The reviewer is right that the open-ended question and the seven everyday situations capture different aspects of mental representations. The idea was to cross-reference the results. The objective of the open question was to analyze the semantic field used spontaneously to answer a question likely to evoke a visual representation (“describe what comes to your mind when you hear the name of a city you have already been to”). We expected that a greater tendency to report visual strategies in everyday life would be accompanied by the use of a lexical field more related to perceptual (visual) aspects of the mental representation.

7) Page 10 reports medians, which is consistent with a non-parametric analysis, but some report of dispersion (e.g. IQR) would also help to describe the groups.

* We now report IQRs (see p.10). Note that we report IQRs as a range between the 1st and 3rd quartiles (rather than as a value) to convey information about the asymmetrical/symmetrical distribution of scores around the median.

---

## [Editor Report · Decision Letter 1]

9 Jul 2021

Thinking in pictures in everyday life situations among autistic adults.

PONE-D-20-39247R1

Dear Dr. Bled,

We’re pleased to inform you that your manuscript has been judged scientifically suitable for publication and will be formally accepted for publication once it meets all outstanding technical requirements.

Kind regards,

Barbara Dritschel, PhD

Academic Editor

PLOS ONE
---

## [Editor Report · Acceptance letter]

14 Jul 2021

PONE-D-20-39247R1 

Thinking in pictures in everyday life situations among autistic adults. 

Dear Dr. Bled:

I'm pleased to inform you that your manuscript has been deemed suitable for publication in PLOS ONE. Congratulations! Your manuscript is now with our production department. 

Kind regards, 

on behalf of

Dr. Barbara Dritschel 

Academic Editor

PLOS ONE